# Changing Prevalence of AIDS and Non-AIDS-Defining Cancers in an Incident Cohort of People Living with HIV over 28 Years

**DOI:** 10.3390/cancers16010070

**Published:** 2023-12-22

**Authors:** Anna Maria Cattelan, Maria Mazzitelli, Nicolò Presa, Claudia Cozzolino, Lolita Sasset, Davide Leoni, Beatrice Bragato, Vincenzo Scaglione, Vincenzo Baldo, Saverio Giuseppe Parisi

**Affiliations:** 1Infectious and Tropical Diseases Unit, Padua University Hospital, 35128 Padua, Italynicolo.presa@aopd.veneto.it (N.P.); lolita.sasset@aopd.veneto.it (L.S.); davide.leoni@aopd.veneto.it (D.L.); beatrice.bragato@aopd.veneto.it (B.B.); vincenzo.scaglione@aopd.veneto.it (V.S.); 2Department of Cardiothoracic and Vascular Sciences and Public Health, Padua University, 35122 Padua, Italy; claudia.cozzolino@studenti.unipd.it (C.C.);; 3Department of Molecular Medicine, Padua University Hospital, 35128 Padua, Italy; saverio.parisi@unipd.it

**Keywords:** HIV, PLWH, cancer, malignancies, epidemiology, AIDS-defining cancer, non-AIDS-defining cancer, tumor

## Abstract

**Simple Summary:**

People living with HIV have an increased risk of developing cancer during their lifetime. Thanks to the introduction and evolution of antiretroviral therapy, cancer phenotypes have changed in this population. Indeed, cancers related to immune suppression (AIDS-related) have decreased, while cancers not related to immune suppression (non-AIDS-related) have progressively increased. In this work, we describe how cancer prevalence has changed in our cohort, confirming data from the literature, in terms of the prevalence of different tumors over time and in terms of patient characteristics. Moreover, to allow a timely diagnosis of non-AIDS-related tumors, it is crucial to promote both adherence to proactive screening and a healthy lifestyle in this population.

**Abstract:**

Background: The introduction and evolution of antiretrovirals has changed the panorama of comorbidities in people living with HIV (PLWH) by reducing the risk of AIDS-defining cancers (ADC). By contrast, due to ageing and persistent inflammation, the prevalence and incidence of non-AIDS-defining cancers have significantly increased. Therefore, we aimed at describing cancer epidemiology in our cohort over 28 years. Methods: We retrospectively included all PLWH in our clinic who ever developed cancers, considering features of ADC and NADC, from January 1996 to March 2023. Demographic, clinical characteristics, and survival were analyzed, comparing three observation periods (1996–2003, 2004–2013, and 2014–2023). Results: A total of 289 PLWH developed 308 cancers over the study period; 77.9% were male, the mean age was 49.6 years (SD 12.2), and 57.4% PLWH developed NADC and 41.5% ADC. Kaposi (21.8%) and non-Hodgkin lymphoma (20.1%) were the most frequent cancers. Age at the time of cancer diagnosis significantly increased over time (41.6 years in the first period vs. 54.4 years in the third period, *p* < 0.001). In the first period compared with the last, a simultaneous diagnosis of HIV infection and cancer occurred in a higher proportion of persons (42.7 vs. 15.3, *p* < 0.001). While viro-immunological control at cancer diagnosis significantly improved over time, the proportions of cancer progression/remission remained stable. Overall survival significantly increased, but this trend was not confirmed for ADC. Conclusions: The probability of survival for ADC did not decrease as significantly as the number of ADC diagnoses over time. By contrast, NADC dramatically increased, in line with epidemiological studies and other literature data. The changing patterns of malignancies from ADC to NADC underline the need for public health interventions and the fostering of screening programs aimed at the prevention and early detection of NADC in PLWH.

## 1. Introduction

The life expectancy of people living with HIV (PLWH) is approaching that of the general population [1] thanks to antiretroviral regimens (triple or dual) that are increasingly powerful and tolerable [2,3]. Therefore, it is expected that by 2030, in Western countries, the majority of PLWH will be over 50 years of age and will therefore present with an increasing prevalence of non-communicable co-morbidities that will influence prognosis and justify choices of or changes in antiretroviral therapy and the need for multidisciplinary approaches [4,5,6]. Indeed, to manage comorbidities and the consequences of polypharmacy, specialized clinics have been implemented.

The increase in mortality and morbidity detected in PLWH is related mostly to the higher incidence of cardiovascular diseases, tumors, neurodegenerative diseases, and endocrine–metabolic dysfunctions [7]. The leading causes of death are currently cardiovascular diseases, metabolic diseases, and cancers [7,8]. Among these, malignancies play a primary role, both for AIDS-defining cancers (ADCs) and for non-AIDS-defining neoplasms (non-AIDS-defining cancers, NADCs). NADCs are now one of the leading causes of mortality among HIV-infected patients [7].

Malignancies, in addition to being the cause of increased mortality, have a strong impact on quality of life, affecting individuals both psychologically and emotionally. Therefore, it is necessary to timely screen and promptly treat people with cancers.

The most important factor related to the increased risk of developing cancer in PLWH is immunosuppression [9]. However, some differences suggest the presence of additional risk factors in PLWH, such as the greater prevalence of traditionally recognized risk factors such as cigarette smoking, alcohol consumption, and the higher prevalence of coinfections with the hepatitis viruses EBV and HPV. Moreover, the mechanisms of carcinogenicity linked to immune activation, chronic inflammation, and immune senescence also characterize this population [10]. Lastly, it is also known that PLWH have a high risk of developing tumors associated with viral co-infections [11].

Thanks to antiretrovirals, the prevalence of ADCs has significantly decreased over time, but, conversely, an alarming and progressive increase in NADCs has been observed [12].

The correlation between immunosuppression and the development of these neoplasms is less evident than in ADCs, and other factors appear to contribute, which differ between the various types of tumors [12]. In general, increases appear to be prevalent for tumors associated with co-infections with oncogenic viruses (anal carcinoma, HL, HCC, leiomyosarcoma, oropharyngeal carcinoma, and nasopharyngeal carcinoma), skin tumors (melanoma and non-melanoma), and tumors associated with smoking (lung cancer). Conversely, colon cancer, prostate cancer, breast cancer, and ovarian cancer do not seem to have a higher incidence in the HIV-positive population than in the general population [12].

This study aims to characterize the evolution of neoplastic pathology in the context of HIV infection from an epidemiological and clinical point of view, comparing three observation periods: the first period (1996–2003) marked by the introduction and initial diffusion of the first combined antiretroviral treatment regimens, and the second (2004–2013) and third (2014–2023) periods characterized by treatment schemes with excellent efficacy and tolerability profiles (the second without the presence of dual therapies and without integrase inhibitor combinations). During these three periods, indeed, the variability of different antiretroviral combinations greatly changed. From 1996–2003 cART was not extensively implemented, and people did not start treatment until CD4+ T cell count showed a significant decrease. In the second period, the use of cART was more prevalent, and during the last period, besides its extensive introduction in clinical practice regardless of immunological levels, integrase inhibitors completely changed the panorama, being characterized by high potency, very few drug interactions, and very high tolerability and safety, especially if compared with boosted protease inhibitors. In this context, the prevalence of people with undetectable HIV RNA has progressively increased over time, lowering their risk of developing AIDS-related cancers directly correlated with immunosuppression.

The main objectives of the study were to evaluate the incidence trend for overall and distinct tumors in ADC and NADC for the whole observation period (i); to evaluate the clinical and viro-immunological characteristics of PLWH with cancers over the three study periods (ii), and to analyze overall and distinct survival by type of tumor, by period of observation, and by course of the disease (iii). Lastly, we calculated disease-free survival (DFS), considering and analyzing the clinical characteristics of people who reported more than one cancer.

## 2. Materials and Methods

This is an observational retrospective cohort study, conducted at the Infectious and Tropical Diseases Unit of Padua University Hospital. The study was conducted in accordance with the Declaration of Helsinki, and it was approved by the Padua Hospital Ethics Committee (protocol code 5285AOP). Participants’ consent was waived per Italian law (AIFA, 31 March 2008). We included all people who were diagnosed with cancer between January 1996 and March 2023 (28 years). The main investigative tools used to identify PLWH diagnosed with cancer were outpatient medical records, day hospital visits and hospitalization, and regional databases including codes related to malignancies (048). This code is assigned by the Italian health system to each person who is diagnosed with cancer and allows them to receive all cancer-related care for free.

We considered all primary cancers occurring after HIV diagnosis or concomitantly with it, as diagnosed by histopathological or radiological examination as required by the diagnostic criteria for each single type of tumor. We also decided to include different tumors diagnosed in the same patients, while tumors occurring before HIV diagnosis were not included. This cohort is to be considered as dynamic over time due to deaths, loss to follow-up, and new entries (incident cohort). Follow-up was calculated both in months and in person–years, from cancer diagnosis to date of disease progression, death, or loss to follow-up or up to 31 May 2023 for the remaining cases.

For each person included, we collected demographics, nationality, date of HIV diagnosis, risk factors for HIV acquisition, date of AIDS diagnosis according to the 1993 CDC criteria [13], nadir CD4+ T cell count, date of diagnosis and typology of the tumor based on the anatomical site of first localization according to the International Classification of Diseases (ICD-10), immuno-virological parameters at the time of diagnosis of the tumor and at the last observation (values of CD4+ T lymphocytes and viral load with limit of detection of 40 HIV-RNA copies per milliliter of plasma), type of antiretroviral at the time of cancer diagnosis, treatment of cancers (classified as surgery/radiotherapy/chemotherapy), and cancer outcome (alive/deceased/lost to follow-up) and relative date and cause of death (distinguished into progression of the tumor and other causes). Tumors were also distinguished into ADCs and NADCs based on the CDC classification [14].

Annual raw incidences were obtained by dividing the number of new cancer diagnoses by the number of people with HIV actively followed up in our clinic. Subjects already presenting a diagnosis of cancer at the beginning of each period were excluded from the denominator. Raw incidence rates (per 1000 PLWH) were averaged over a 4-year period. Trends were also tested with simple linear regression.

A descriptive analysis was performed on the entire sample, representing categorical variables as frequencies and (column) proportions while summarizing continuous numerical variables with mean values, medians, standard deviations (SD), and interquartile ranges (IQR). Bivariate analyses were conducted to investigate possible differences in demographic and clinical characteristics among ADCs vs. NADCs and among HIV subjects diagnosed with cancer in three distinct decades: 1996–2003, 2004–2013, and 2014–2023. The Chi-squared test, or, alternatively, Fisher’s exact test (in the presence of an expected frequency lower than five), was applied to address differences in the distribution of the categorical variables among the groups. The Kruskal–Wallis one-way analysis of variance (ANOVA) was performed to quantify the group effect on numerical features (it was not possible to adopt the standard parametric ANOVA, as the Shapiro–Wilk test highlighted significant departures from the normal distribution).

The two-way ANOVA and Mann–Whitney–Wilcoxon test were adopted to compare numerical variables, particularly time from HIV to cancer diagnosis, between ACD and NADC over the three decades.

Follow-up time was calculated in terms of person-years, estimating the actual time-at-risk—in years—that all participants contributed to the study.

Long-term (15-year) overall and disease-free survival (DFS) were analyzed stratifying by decade of diagnosis and by tumor type (ADC versus non-ADC) using Kaplan–Meier curves. DFS was defined as the time from diagnosis to recurrence of tumor, death, or loss to follow-up. Significant variations in long- (15-year) and short-term (2-year) survival by group were determined with the log-rank test. Finally, Cox regression was used to assess the hazard ratio (HR) for disease-free survival for a range of factors. HR were estimated both with univariate and multivariable models, adjusting by demographics, decade, and tumor type. The regression analysis was also repeated separately for each different decade. To avoid bias when comparing cases with recent (2014–2023) rather than older (1996–2013) cancer diagnoses, follow-up time was truncated to only two years. In addition, only primary cancers were considered for the overall survival analysis.

Results were deemed as statistically significant at *p* < 0.05. All data analyses and visualization were conducted in R 4.0.4.

## 3. Results

Between 1 January 1996, and 31 March 2023, 289 PLWH developed 308 cancers during a total of 1731.7 person-years of follow-up. The overall characteristics of the study population are reported in Table 1 (please also refer to Appendix A for statistics by decade of diagnosis). The two-hundred and twenty-five participants (77.9%) were male and predominantly (91%) Italian; their median age was 49 years (IQR: 41–58); and the transmission route was mostly unprotected sex between men (50.9%), followed by unprotected heterosexual intercourse (26%) and intravenous drug use (22.1%).

At HIV diagnosis and at first cancer presentation, the median CD4+ T cell count was 147 cells/μL (IQR: 70–290 and 266 cells/μL (IQR: 120–540), respectively.

At first cancer diagnosis, 188 (65.1%) individuals were on antiretroviral therapy and 141 (48.8%) had a viral load < 50 copies/mL. During the study period, 75 (26%), 90 (31.1%), and 124 (42.9%) primary tumors were diagnosed in the first (1996–2003), the second (2004–2013) and the third (2014–2023) decade, respectively. Most of PLWH first developed NADC (166/289, 57.4%), while 120 (41.5%) developed ADC. Data on tumor type were missing for the remaining three PLWH. Approximately one quarter of the individuals (72, 24.9%) received HIV and cancer diagnosis simultaneously; these latter cases were mainly ADC (59.2% vs. 0.6%; *p* < 0.0001). In total, 20 out of 289 (6.9%) PLWH developed more than one cancer (6.6% two cancers and 0.3% three cancers).

The most common cancer was Kaposi sarcoma (21.8%), followed by non-Hodgkin lymphoma (20.1%), hepatocellular carcinoma (7.3%), anal cancer (6.6%), and non-melanoma skin cancer (5.5%). We observed 28 equally divided cases (4.8% and 4.8%) of breast cancer and neoplasms of the oral cavity, pharynx, and salivary glands, and a similar pattern of prevalence for cervical (12, 4.2%) and prostate (12, 3.9%) cancer. Nineteen persons (6.6%) were diagnosed with cancers we classified as “other”. We observed one ovarian cancer, one pancreatic cancer, two kidney cancers, two stomach cancers, two testicular cancers, two thyroid cancers, two vulvar cancers, three osteosarcomas, and four cases of cholangiocarcinoma.

More than 90% of PLWH received a specific anti-tumor treatment during the study period: chemotherapy, surgery, and radiation therapy in 47.9%, 40.5%, and 17.7% of cases, respectively. Chemotherapy was significantly more frequently used in ADC than in NADC (58% vs. 40.6%; *p* = 0.0041), while a surgical approach was more prevalent in NADC than in ADC (66.7% vs. 4.2%; *p* < 0.0001). The overall number of deaths was 85 (29.4%); 39 occurred in the ADC group and 46 in the NADC group.

The number of ADCs progressively decreased over the three decades of the study, while the number of NADCs progressively increased (*p* < 0.0001, Figure 1).

Moreover, the time from HIV diagnosis to cancer development progressively increased over time, especially among NADCs (median values from 1.8 years to 15.5, Appendix A), with statistically significant differences in tumor types for the last two decades (*p* < 0.0001, Figure 2).

Median age at the time of cancer diagnosis significantly increased over time (41.6 years in the first period vs. 54.4 years in the third period, *p* < 0.001, Appendix A). No differences in sex distribution were detected during the three decades, with a steadily higher prevalence in men (86.7% to 74.2%, *p* = 0.0994). In the first period, compared with the last, simultaneous diagnosis of HIV infection and cancer occurred in a higher proportion of persons (42.7% vs. 15.3%, *p* < 0.001). Since no antiretrovirals or sub-optimal antiretroviral regimens were available during the first decade, the proportion of people who were undetectable (i.e., HIV RNA < 50 copies/mL) at the time of cancer diagnosis was significantly higher during the third decade compared with the first (74.2% vs. 8.0%, *p* < 0.001). Similarly, the number of people with both CD4+ T cell count <500 cell/mm^3^ and <200 cell/mm^3^ was significantly higher during the first decade compared with the third (*p* < 0.0001). While viro-immunological control at cancer diagnosis significantly improved over time, the proportions of cancer progression and remission remained stable (*p* = 0.0515). Complete remission was obtained in 60% of tumors without significant differences between ADC and NADC (65.5% vs. 55.8%, *p* = 0.143, Table 2).

Overall, estimated 2-year overall survival and DFS significantly increased during the three study periods. This trend was not confirmed for long-term outcomes or in the ADC group (Figure 3 and Figure 4). From the univariate Cox regression, diagnosis in the more recent decades 2004–2013 or 2014–2023 (HR = 0.59, 0.46 reference 1996–2003), higher CD4 + T cell count at HIV diagnosis (HR = 0.99), undetectable viremia (HR = 0.51), CD4+ T cell count >500 per mcL (HR = 0.42), surgery (HR = 0.31), and anticancer treatment (HR = 0.1) all turned out to be statistically significant (*p* < 0.05) protective factors for DFS. T CD4 < 200 per mcL (HR = 2.06) and undergoing chemotherapy (HR = 1.87) were found instead to be statistically significant (*p* < 0.05) risk factors. In contrast, tumor type (ADC versus NADC) and the other variables here described were not found to have a significant impact on short-term DFS (HR~1, *p* > 0.05).

The multivariable analysis only confirmed CD4+ T cell count at HIV diagnosis (HR = 0.99), diagnosis in the second decade (HR = 0.33), and surgical (HR = 0.03) and chemo (HR = 4.01) treatment as factors significantly (*p* < 0.05) associated with DFS.

## 4. Discussion

In the era of combination antiretroviral therapy (cART), there has been a remarkable shift in the landscape of HIV infection. What was once a relatively acute and life-threatening condition has evolved into a chronic illness characterized by the presence of various comorbidities, including NADCs [1,15]. In our study, we assessed 28-year trends in the frequency and spectrum of tumors in PLWH. Participant enrollment started in 1996 when highly active antiretroviral therapy (HAART) had just been introduced into clinical practice and ended in March 2023. Over the study period, a significant increase in the prevalence of NADCs, especially those unrelated to viral infections, was observed. Interestingly, when comparing the initial period of the study with the last one, we observed an inverse and mirrored distribution of ADC and NADC: in the first period, 77% of cases were ADC, whereas in the third period this figure decreased to 22%. Conversely, the prevalence of NADC cases increased from 22% in the first period to 77% in the third period. However, in last two decades, the trend was no longer linear, showing a peak for NADC in 2012–2015 and a recovery of ADC in the last period. It is possible to assume that the peak for NADC is due to the implementation of cancer screening that began in those years, and that the recovery of ADC in the last period is linked to the increase in late HIV presentation in our cohort of PLWH.

Indeed, the etiology of NADCs in PLWH is multifaceted and may be linked to a confluence of factors, including the aging population of people living with HIV.

This aligns with the evolving epidemiological patterns and population dynamics of PLWH in the United States, as outlined by Shiels and colleagues [16]. They utilized a model to project anticipated cancer incidence (total case numbers) among people living with HIV in 2020 and 2030. This suggests that a growing prevalence of non-AIDS-defining cancers not linked to viral factors is projected to persist among PLWH in the coming decades.

Also, of note in our study, age at the time of cancer diagnosis significantly increased over time (41.6 years in the first period vs. 54.6 years in the third period, *p* < 0.001). This shift reflects the effectiveness of cART, enabling PLWH to live longer [17]. However, this extended lifespan also makes them more susceptible to various age-related health conditions, including cancer. Nevertheless, it is worth noting that the median age remains lower than the median age for cancer development observed in the population without HIV [9,18,19]. This suggests that, in PLWH, other factors, such as co-infections with oncogenic viruses, heightened rates of tobacco and substance use, medication usage, HIV-associated metabolic disturbances, and immune inflammation, may all contribute to this clinical correlation. For instance, cigarette smoking is more prevalent among PLWH compared with people without HIV in many countries, likely contributing to an elevated risk of several tumor types [20]. Additionally, the modulation of oncogenic viral co-infections in PLWH, such as the Epstein–Barr virus, human papillomavirus, hepatitis B, and hepatitis C, is associated with increased risks of virally mediated cancers, including anal and liver cancers [21,22]. In this study, it is interesting to note that lung, anal, and liver tumors collectively accounted for 18% of all recorded cancers. In addition, in our cohort of PLWH, concurrent diagnosis of HIV and cancer progressively decreased over time and was statistically significantly different between the first and third period of observation. This is in line with the SMART and TEMPRANO studies, in which they have shown that starting ART early, regardless of CD4+ T cell count, can lead to improved health outcomes, including a decreased risk of opportunistic infections and, to some extent, a reduction in overall mortality [23,24].

Furthermore, it has been reported that HIV disease severity has also been implicated in an increased risk of various NADCs. HIV viral suppression was broadly linked to a decreased risk of NADCs in a national study of US veterans [25]. Similarly, the presence of an AIDS diagnosis, a CD4+ T cell count below 200 cells per μL after 6 months on antiretroviral therapy (ART), a low nadir CD4+ T cell count, and prolonged periods of low CD4+ T cell count have been associated with elevated standardized incidence ratios (SIRs) for some non-AIDS-defining cancers in population-based studies [26]. In addition to the decline in CD4+ T-cell immunity caused by HIV infection, a decreased CD4:CD8 ratio has also shown its potential as a predictive biomarker for non-AIDS-defining cancer risk within large HIV study cohorts [27]. Unfortunately, we did not collect the CD4:CD8 ratio in our patients. However, the results obtained in our cohort of people living with HIV (PLWH), which demonstrate a progressive increase in CD4+ T cell count at the time of tumor diagnosis, a progressively higher nadir CD4+ T cell number, and a greater number of individuals treated with effective antiretroviral therapy, may suggest that, especially in the last decade of the study, factors related to immune aging or coinfections are more implicated in NADC development than in the severity of HIV disease itself.

Overall, in our study, we found that the proportions of cancer progression and remission did not significantly vary over time, indicating a stable balance between ADC and NADC trajectories throughout the extended follow-up period. Over the three decades of our study, we observed a marked decrease in AIDS-defining cancers (ADCs) and a significant increase in non-AIDS-defining cancers (NADCs). DFS improved significantly for NADCs, while it remained constant for ADCs. Furthermore, contrary to expectations, disease-free survival in the two-year follow-up of ADCs was observed to be lower, although not statistically significantly, during the period from 2014 to 2023 compared with the period from 2004 to 2013. We can hypothesize that ADCs were not only diagnosed in very advanced HIV infection but also frequently associated with other opportunistic infections. However, no further investigation was conducted in this regard.

Interestingly, in the adjusted analysis, a specific HIV disease marker, the nadir CD4+ T cell count, remained the unique factor inversely correlated with disease-free survival. This observation contributes to the ongoing discussion on the complex relationship between the potential oncogenic role of HIV, conventional cancer risks, and the protective role of antiretroviral therapy in cancer development among PLWH. Regarding antiretroviral therapy, some researchers have speculated that a potential involvement of integrase inhibitors and tenofovir alafenamide in the carcinogenic process is associated with the BMI increase triggered by these medications. A prior study conducted in EuroSIDA observed that the percentage of individuals classified as overweight or obese rose during the study period with a median follow-up of four years [28]. This elevation in weight did not result in a heightened risk of malignancies, although it is possible that a four-year duration may be insufficient for cancer to manifest [28]. We believe that, to date, the benefits of antiretroviral therapy far outweigh the potential drawbacks, and any side effects can be addressed mainly through lifestyle changes. Indeed, more research is needed to better understand the implications of the association of weight gain with the new antiretroviral regimens and the trends in BMI-related cancers.

In our study, we observed a notable upward trajectory in the prevalence of cancer cases, particularly those unrelated to viral infections, during the long study period ranging from 1996 to 2023. The increasing tumor trend was similar between men and women, providing further evidence that no gender differences can be observed in the risk of developing cancer in PLWH [19].

Not surprisingly, aging was significantly linked to NADCs. Due to the effectiveness of cART, PLWH are now living longer, which in turn increases their susceptibility to various age-related health conditions, including cancer. In fact, previous studies revealed that the higher incidence of NADCs during the cART era can largely be attributed to aging in PLWH [29,30], and the number of cancers is expected to continue to rise. Conversely, a clear association with immunodeficiency has only been established for cancers that are AIDS-defining and those that are non-AIDS-defining but have a known infectious cause.

In our study, among people diagnosed with NADCs, it was observed that 143 out of 166 individuals (86.1%) had a prior history of antiretroviral treatment and 33 (21.3%) had CD4 counts of less than 200 cells per microliter at the time of NADC diagnosis. To gain a more comprehensive understanding of the impact of the severity and duration of immunosuppression on the development of NADCs, larger prospective studies are warranted. These studies would need to consider the time-dependent aspects of immunosuppression to provide clearer insights into this complex relationship.

We should acknowledge that the retrospective nature of our study, the relatively small number of NADC events, and the difference in the frequencies of CD4+ T cell and HIV-RNA- measurements during the follow-up period might have limited the strength of our analysis.

## 5. Conclusions

In conclusion, the findings of our study underscore the importance of continued research, prevention, and early detection efforts in the context of HIV and associated cancer risk factors, as well as of prevalence and incidence studies. Effective strategies for reducing the risk of these cancers include lifestyle improvements, limiting alcohol consumption, HPV vaccination, smoking cessation programs, and the management of HBV/HCV co-infections, but also fostering the uptake of active surveillance screening programs, such as cytology for cervical lesions, HPV testing and mammograms for women, screening for lung cancer with low-dose computerized tomography in heavy smokers, anal cancer screening, and colorectal cancer screening. Monitoring and addressing the unique healthcare needs of individuals living with HIV is crucial for cancer prevention and control in this population.

## Figures and Tables

**Figure 1 cancers-16-00070-f001:**
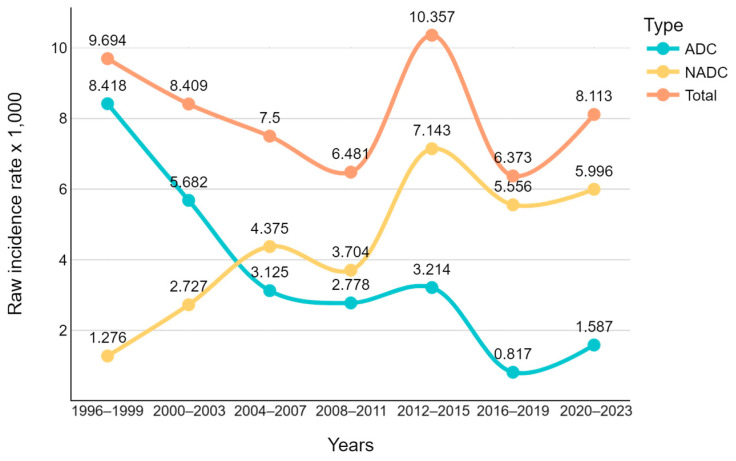
Annual raw incidence per 1000 PLWH of ADC and NADC over time.

**Figure 2 cancers-16-00070-f002:**
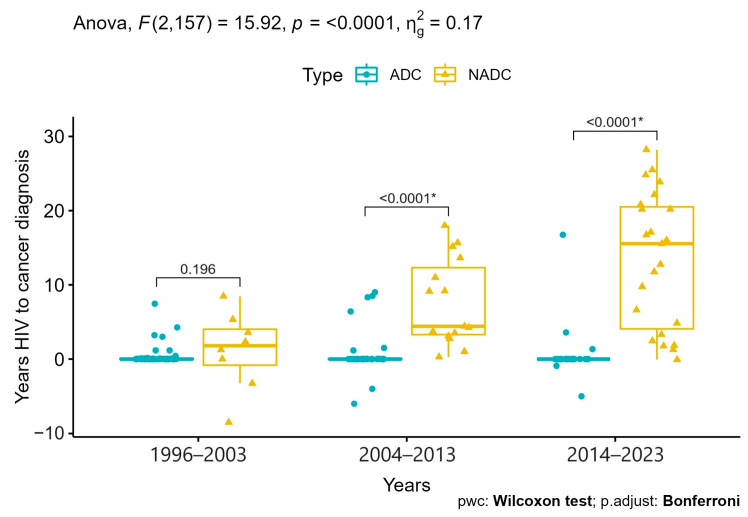
Differences in time from HIV to cancer diagnosis between ACD and NADC over the three decades. Pairwise comparison *p*-values marked with * fall below the significance level under Bonferroni correction.

**Figure 3 cancers-16-00070-f003:**
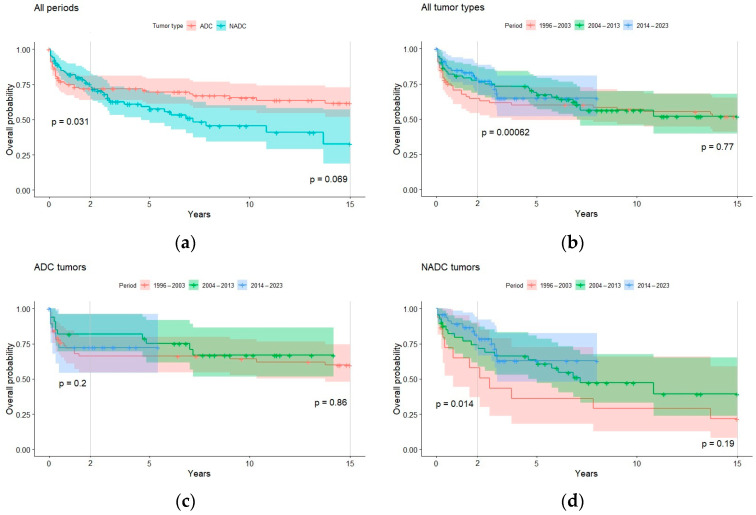
Survival probabilities for all periods (**a**), all tumor types (**b**), ADC (**c**), and NADC (**d**). Significance values for variations in long- (15-year) and short-term (2-year) survival by group were determined with the log-rank test.

**Figure 4 cancers-16-00070-f004:**
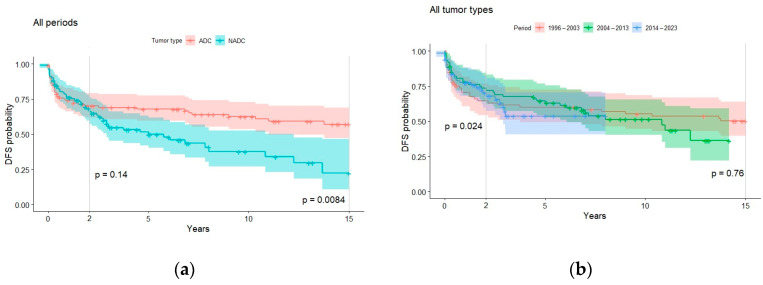
Disease-free survival probabilities for all periods (**a**), all tumor types (**b**), ADC (**c**), and NADC (**d**). Significance values for variations in long- (15-year) and short-term (2-year) survival by group were determined with the log-rank test.

**Table 1 cancers-16-00070-t001:** Characteristics of the study population, overall and by ACD and NADC population (all proportions are intended as column percentages).

Characteristics		Overall *N* = 289	ADC *N* = 120	NADC *N* = 166	*p*-Value
Age (at cancer diagnosis), years, median (IQR)		49 (41–58)	43.5 (36.8–50)	53 (46–61)	<0.0001 ^k^
Sex, male, *N* (%)		225 (77.9)	107 (89.2)	116 (69.9)	0.0001 ^c^
Italian nationality, n (%)		263 (91)	107 (89.2)	154 (92.8)	0.2869 ^c^
Risk factor, n (%)	Blood product	2 (0.7)	0 (0)	2 (1.2)	<0.0001 ^f^
	Unprotected heterosexual sex	75 (26)	22 (18.3)	52 (31.3)	
	Unprotected sex between men	147 (50.9)	82 (68.3)	63 (38)	
	Intravenous drug use	64 (22.1)	16 (13.3)	48 (28.9)	
	Vertical transmission	1 (0.3)	0 (0)	1 (0.6)	
HIV diagnosis coincides with tumor diagnosis, n (%)		72 (24.9)	71 (59.2)	1 (0.6)	<0.0001 ^f^
Years from HIV to cancer diagnosis, median (IQR)		0 (0–3.2)	0 (0–0)	7.5 (2.5–15.9)	<0.0001 ^k^
CD4 + T cell count at HIV diagnosis, cell/mm^3^, median (IQR)		147 (70–290)	100 (40–206)	190 (90–318)	<0.0001 ^k^
CD4 + T cell count at cancer diagnosis, cell/mm^3^, median (IQR)		266 (120–540)	130 (45–246)	469 (242–720)	<0.0001 ^k^
CD4 + T cell count at cancer diagnosis <200 CD4/mcL, n (%)		111 (40.7)	77 (67)	33 (21.3)	<0.0001 ^c^
CD4 + T cell count at cancer diagnosis >500 CD4/mcL, n (%)		80 (29.3)	10 (8.7)	67 (43.2)	<0.0001 ^c^
Undetectable viremia (<50 cp/mL), at cancer diagnosis n (%)		141 (48.8)	19 (15.8)	120 (72.3)	<0.0001 ^c^
IHAART exposure at cancer diagnosis, n (%)		188 (65.1)	42 (35)	143 (86.1)	<0.0001 ^c^
Decade of cancer diagnosis, n (%)	1996–2003	75 (26)	58 (48.3)	17 (10.2)	<0.0001 ^c^
	2004–2013	90 (31.1)	35 (29.2)	55 (33.1)	
	2014–2023	124 (42.9)	27 (22.5)	94 (56.6)	
Tumor, n (%)	Kaposi	63 (21.8)	63 (52.5)	0 (0)	<0.0001 ^f^
	NHL	58 (20.1)	57 (47.5)	1 (0.6)	
	HCC	21 (7.3)	0 (0)	21 (12.7)	
	Anal cancer	19 (6.6)	0 (0)	18 (10.8)	
	Non-melanoma skin cancer	16 (5.5)	0 (0)	16 (9.6)	
	Neoplasms of the oral cavity, pharynx, and salivary glands	14 (4.8)	0 (0)	14 (8.4)	
	Breast cancer	14 (4.8)	0 (0)	14 (8.4)	
	Prostate cancer	12 (4.2)	0 (0)	12 (7.2)	
	Cervical cancer	12 (4.2)	0 (0)	11 (6.6)	
	Colorectal cancer	11 (3.8)	0 (0)	11 (6.6)	
	Lung cancer	9 (3.1)	0 (0)	8 (4.8)	
	Malignant lymphoproliferative diseases	8 (2.8)	0 (0)	8 (4.8)	
	Bladder cancer	7 (2.4)	0 (0)	7 (4.2)	
	Melanoma	6 (2.1)	0 (0)	6 (3.6)	
	Other	19 (6.6)	0 (0)	19 (11.4)	

Statistical methods: c. Chi-squared test, f. Fisher’s exact test, k. Kruskal–Wallis ANOVA.

**Table 2 cancers-16-00070-t002:** Cancer treatments and outcomes, overall and by ACD and NADC population (all proportions are intended as column percentages).

Characteristics		Overall *N* = 289	ADC *N* = 120	NADC *N* = 166	*p*-Value
Anticancer treatment, *N* (%)		273 (94.8)	114 (95.8)	156 (94)	0.4968 ^c^
Chemotherapy, n (%)		135 (47.9)	69 (58)	65 (40.6)	0.0041 ^c^
Radiotherapy, n (%)		50 (17.7)	18 (15.1)	32 (20)	0.2938 ^c^
Surgical treatment, n (%)		115 (40.5)	5 (4.2)	108 (66.7)	<0.0001 ^c^
Tumor outcome, n (%)					0.1430 ^c^
	Progression	71 (24.9)	23 (19.3)	48 (29.4)	
	Complete remission	171 (60)	78 (65.5)	91 (55.8)	
	Partial remission	43 (15.1)	18 (15.1)	24 (14.7)	
	Deaths	85 (29.4)	39 (32.5)	46 (27.7)	
2-year mortality, n (%)		55 (19.1)	30 (25)	25 (15.2)	0.0375 ^c^
10-year mortality, n (%)		78 (27.1)	35 (29.2)	43 (26.1)	0.5615 ^c^
Follow-up, person-years		1731.7	899.9	825.4	-

Statistical method: c. Chi-squared test.

## Data Availability

The data that support the findings of this study are available on request from the corresponding author. The data are not publicly available due to privacy or ethical restrictions.

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
