# Peer review of "Changing Prevalence of AIDS and Non-AIDS-Defining Cancers in an Incident Cohort of People Living with HIV over 28 Years"

_cancers, 2023, doi:10.3390/cancers16010070_

Round 1

Reviewer 1 Report

Comments and Suggestions for Authors

Thank you very much for the opportunity to review your research. This manuscript presents a retrospective cohort study spanning 28 years, providing a comprehensive overview of trends and changes in the epidemiology of cancer in people living with HIV (PLWH) over an extended period.

The study not only explores the overall cancer incidence but also distinguishes between AIDS-defining cancers (ADC) and non-AIDS-defining cancers (NADC). This differentiation is extremely interesting and allows for a nuanced understanding of changing patterns in different types of malignancies.

Among the strengths of the study is the inclusion of demographic, clinical, and survival data, enabling a thorough examination of factors influencing cancer development and outcomes among PLWH. The comparison between three distinct observation periods adds depth to the evaluation.

To facilitate the publication of the manuscript, I suggest some minor changes:

1. The title could be more specific about the main findings or contributions of the study, providing a clearer indication of the focus for potential readers.

2. The abstract is incomplete, with a brief and non-explanatory background. Overall, the text does not provide a concise summary of the main conclusions, making it difficult for readers to quickly understand the key results and implications of the study.

3. The manuscript lacks clearer and more explanatory visual resources, which could enhance the presentation of key demographic and clinical features, making it more accessible to readers.

4. The introduction is still incomplete, lacking some critical information, such as specific details about antiretroviral regimens used in different periods, which may be relevant for understanding changes in cancer incidence.

5. *The survival analysis truncates the study for the first two years of follow-up for cases with a recent cancer diagnosis (2014-2023), potentially limiting the generalization of long-term survival trends. Please review.

6. Some statistical results are mentioned in the narrative without corresponding p-values, making it difficult to assess the statistical significance of certain findings. I may not have understood this correctly, so I ask for a better explanation of this limitation.

7. While the study identifies trends, a more in-depth discussion and interpretation of the implications of these trends for clinical practice or public health interventions could strengthen the manuscript.

Please address these points in your revision. Acknowledging these limitations and improving the clarity of the presentation would enhance its impact and contribution to the field.

Author Response

Reviewer #1

Thank you very much for the opportunity to review your research. This manuscript presents a retrospective cohort study spanning 28 years, providing a comprehensive overview of trends and changes in the epidemiology of cancer in people living with HIV (PLWH) over an extended period. The study not only explores the overall cancer incidence but also distinguishes between AIDS-defining cancers (ADC) and non-AIDS-defining cancers (NADC). This differentiation is extremely interesting and allows for a nuanced understanding of changing patterns in different types of malignancies. Among the strengths of the study is the inclusion of demographic, clinical, and survival data, enabling a thorough examination of factors influencing cancer development and outcomes among PLWH. The comparison between three distinct observation periods adds depth to the evaluation.

AR: Dear Reviewer, thanks for appreciating our work and for the time you spent in the revision process. We tried to clear all the issues you raised, hoping that now the manuscript is suitable for publication.

To facilitate the publication of the manuscript, I suggest some minor changes:

  1. The title could be more specific about the main findings or contributions of the study, providing a clearer indication of the focus for potential readers.

AR: Dear Reviewer, we changed the title trying to better highlight what we explored in our work.

  1. The abstract is incomplete, with a brief and non-explanatory background. Overall, the text does not provide a concise summary of the main conclusions, making it difficult for readers to quickly understand the key results and implications of the study.

AR: Thanks for this comment, the abstract was amended and better structured, especially the background section.

  1. The manuscript lacks clearer and more explanatory visual resources, which could enhance the presentation of key demographic and clinical features, making it more accessible to readers.

AR: Dear reviewer, we added many materials to the manuscript (also as supplementary), which hopefully will support the readers.

  1. The introduction is still incomplete, lacking some critical information, such as specific details about antiretroviral regimens used in different periods, which may be relevant for understanding changes in cancer incidence.

AR: Thanks for this observation. We added a further paragraph discussing the differences among the three period.

  1. *The survival analysis truncates the study for the first two years of follow-up for cases with a recent cancer diagnosis (2014-2023), potentially limiting the generalization of long-term survival trends. Please review.

AR: Thank you for bringing out this aspect. Since our incident court has very heterogeneous follow-up times, a drastic crop on observation time had been adopted to ensure a fair comparison between decades. To better investigate long-term survival, fully exploiting our 30-year cohort, we recalculated the Kaplan Mayer curves and the log-rank test up to 15 years (please see the new Figures 3 and 4. However, the Cox regressions were kept with the shorter-term outcome to avoid biases.

  1. Some statistical results are mentioned in the narrative without corresponding p-values, making it difficult to assess the statistical significance of certain findings. I may not have understood this correctly, so I ask for a better explanation of this limitation.

AR: Thank you for the correction, we carefully reviewed the results so that they were always rigorously supported by statistical evidence. We have also included further data as supplementary materials to support all the data presented.

  1. While the study identifies trends, a more in-depth discussion and interpretation of the implications of these trends for clinical practice or public health interventions could strengthen the manuscript. Please address these points in your revision. Acknowledging these limitations and improving the clarity of the presentation would enhance its impact and contribution to the field.

AR: Please note that some points of the discussion which were repeated were removed. We added a proper conclusion trying to foster the idea that especially timely diagnosis and screening uptake are crucial in our population. Limitations are stated into discussion section.

Reviewer 2 Report

Comments and Suggestions for Authors

The authors conducted a retrospective study using medical record of HIV patients at Padua University Hospital in Italy to assess cancer incidence trend, clinical and viro-immunological characteristics of PLHIV who developed cancer, overall survival and disease-free survival. The authors will need to better align the objectives, methods and results (text, tables and figures); and the results in the text to the tables and graphs presented. I would also suggest the authors to enhance the interpretation of statistical results. In addition, consider including the demographic and clinical characteristic by study periods defined since cancer development are associated with the patients’ characteristics and cancer treatment options changed overtime. It will also be helpful to discuss whether the results found from Padua University can be generalize to PLHIV in Italy or not since cancer development is associated with geographic and/or environmental factors. Other suggestions are listed below.

Line 22, Please clarify, age here is mean or median. The author stated that age at the time of cancer diagnosis significantly increase over time. What is median\mean age of PLHIV at the time of HIV diagnosed? Was the trend similar or not? 

Line 97, is “48” an ICD 10 code?

Line 98-101, Please clarify how cancer incidence were calculated? Was it based on the number of PLHIV diagnosed with cancer divided by PLHIV at the time period? Were the subjects who developed cancer prior to HIV diagnosis include in the denominator or not in calculating cancer incidence?

Line 120-122, If I am not miss reading, the results in comparing demographic and clinical characteristics of cancer patients by study periods were not presented in the Table 1.

Line 125-128, Please clarify the Kruskal-Wallis one-way analysis of variance was used for what variables. Was this refer to the results in Table1?

Line 135-136, Please clarify what bias that the authors were referred to.  It is not clear why only the first two years for the period of 2014-2023 were used?

Line 142, Bonferroni adjustment was mentioned in Figure 2. The significant level should be different.

Line 144, Table 1, Were the percentages presented here column percentages?

Line 146, Please clarify the total follow-up year. This might be a typo.

Line 155, Was “most people” here referred to HIV infected patients?

Line 156, By reading the sentence, I think the authors referred to the row percentage, but the column percentage was presented here.  Please check.

Line 173-174, Please clarify in the method how the incidence was calculated in the method section. In the method section, the authors defined 3 study periods. Yet, there are 7 different time points presented Figure 1. Some of time point were in between of the study periods that the author defined. What test was used to assess the trend and estimate the p-value?

If I am not miss reading, the interpretation also is questionable. The incidence of ADC dropped from 1996-2011, increased slightly in 2012-2015 and dropped again in 2016-2019.  I am also wondering why the trends of ADC and NADC become more alike after 2008?

Line 188, Consider presenting the results in a table.  What does p=n.s. and **** mean? (Figure 2)

Line, 202-212, The authors will need to better interpret the results using hazard ratios.

Line 235, Based on Figure 1, there is an opposite trend for ADC and NADC from 1996 to 2007, after that, the trends were similar even though the scale of change was not the same.

For Figure 4, Why were the areas of color different? What does it mean?

Author Response

Reviewer #2

The authors conducted a retrospective study using medical record of HIV patients at Padua University Hospital in Italy to assess cancer incidence trend, clinical and viro-immunological characteristics of PLHIV who developed cancer, overall survival, and disease-free survival. The authors will need to better align the objectives, methods, and results (text, tables and figures); and the results in the text to the tables and graphs presented. I would also suggest the authors to enhance the interpretation of statistical results. In addition, consider including the demographic and clinical characteristic by study periods defined since cancer development are associated with the patients’ characteristics and cancer treatment options changed overtime. It will also be helpful to discuss whether the results found from Padua University can be generalize to PLHIV in Italy or not since cancer development is associated with geographic and/or environmental factors. Other suggestions are listed below.

AR: Dear Reviewer, thanks for appreciating our work and for the time you spent in the revision process. We tried to clear all the issues you raised, hoping that now the manuscript is suitable for publication.

Line 22, Please clarify, age here is mean or median. The author stated that age at the time of cancer diagnosis significantly increase over time. What is median\mean age of PLHIV at the time of HIV diagnosed? Was the trend similar or not?  

AR: We specified as per your request. During the first period median age living with HIV was shorter than that in the last period and people with cancer diagnosis were more likely to be younger. 

Line 97, is “48” an ICD 10 code?

AR: This code is assigned by the Italian health system to each person who is diagnosed with cancer and that allows to receive all the cancer-related care for free. This is now specified into methods.

Line 98-101, Please clarify how cancer incidence were calculated? Was it based on the number of PLHIV diagnosed with cancer divided by PLHIV at the time period? Were the subjects who developed cancer prior to HIV diagnosis include in the denominator or not in calculating cancer incidence?

AR: Thanks, this is now better specified in the methods (see lines 127-130). Annual raw incidences were obtained dividing the number of new cancer diagnosis by the number of people with HIV actively followed-up in our clinic. Subjects already presenting a diagnosis of cancer at the beginning of each period were excluded from the denominator. Raw incidence rates (per 1,000) were averaged over a 4-year period.

Line 120-122, If I am not miss reading, the results in comparing demographic and clinical characteristics of cancer patients by study periods were not presented in the Table 1.

AR: you are right, Table 1 shows characteristics of the study population, overall and by ACD and NADC population. We have now included supplementary materials to support all the data presented.

Line 125-128, Please clarify the Kruskal-Wallis one-way analysis of variance was used for what variables. Was this refer to the results in Table1?

AR: As specified in the methods, the Kruskal-Wallis one-way ANOVA was used for bivariate analysis of continuous numerical variables (results in Table 1). For categorical ones, Chi square and Fisher's tests were used instead. We now clearly state the statistical test for each variable by adding a footnote in Table 1. Thank you for the observation.

Line 135-136, Please clarify what bias that the authors were referred to.  It is not clear why only the first two years for the period of 2014-2023 were used?

AR: Given the heterogeneity of follow-up time in our dynamic cohort, if we had not truncated survival analyses at the observed minimum, we might have obtained meaningless results. Significant associations with some variables might in fact only be due to differences in subjects observed longer than those newly enrolled (or vice versa), rather than due to real effect of certain risk/protective factors on survival. This is especially important when we want to investigate the relationship between survival and decade of diagnosis. In order not to limit our study too much and better exploit our 30-year court, we decided to do truncation only for Log-rank test and Cox regression, not for KM curves. Please see new Figures 3 and 4.

Line 142, Bonferroni adjustment was mentioned in Figure 2. The significant level should be different.

AR: As you rightly note, the Bonferroni adjustment in pairwise comparisons involves a different level from the usual 5%. For this reason, we originally expressed significance directly with standard symbols: ns (p ≥ adj. level), * (p < adj. level), ** (p << adj. level), etc. We have now better detailed significance in Figure 2. Thank you.

Line 144, Table 1, Were the percentages presented here column percentages?

AR: Table 1 shows the results of the bivariate analysis, whose purpose was to determine the differences in the distribution of various factors in the subgroups of PLWH with ADC versus non-ADC. For this reason, column percentages are most appropriate. For clarity we have specified this both in the methods and in the caption.

Line 146, Please clarify the total follow-up year. This might be a typo.

AR: “person-years” unit is correct. It indicates the total follow-up calculated in terms of person-time. Person-time is a standard epidemiological estimate of the actual time-at-risk – here in years – that all participants contributed to a study. For the sake of clarity, we have also defined it in the methods.

Line 155, Was “most people” here referred to HIV infected patients?

  1. Yes, correct. We have rephrased the sentence to improve comprehension. Thank you for the suggestion.

Line 156, By reading the sentence, I think the authors referred to the row percentage, but the column percentage was presented here.  Please check.

AR: The percentages indicate the prevalence of ADC and non-ADC cancers in the whole cohort. We have rephrased the sentence to improve comprehension. Thank you for the suggestion.

Line 173-174, Please clarify in the method how the incidence was calculated in the method section. In the method section, the authors defined 3 study periods. Yet, there are 7 different time points presented Figure 1. Some of time point were in between of the study periods that the author defined. What test was used to assess the trend and estimate the p-value?

AR: Thank you, as above, incidence calculation is now better clarified in the methods section. Four-year averaging was adopted i) to reduce noise in the graphical representation; ii) due to lack of accurate year-by-year size of PLWH followed in our center. The trend was instead tested with a simple linear regression since there were only few points (see lines 127-131).

If I am not miss reading, the interpretation also is questionable. The incidence of ADC dropped from 1996-2011, increased slightly in 2012-2015 and dropped again in 2016-2019.  I am also wondering why the trends of ADC and NADC become more alike after 2008?

AR: We agree with you, hence we preferred to limit ourselves to describing the general trend of our observational time. Unfortunately, we are unable to interpret the fluctuations in the years after 2008. Possible explanations could be related to the evolution of the characteristics and size of our PLWH court (shared denominator) during these 30 years, especially after COVID-19 pandemic. This would require further ad-hoc studies.

Line 188, Consider presenting the results in a table.  What does p=n.s. and **** mean? (Figure 2)

AR: Thank you for the correction. We have replaced ns with the precise value of p, and we have clarified pairwise comparison test significance in Figure 2. We have also included supplementary materials to support all the data presented.

Line, 202-212, The authors will need to better interpret the results using hazard ratios.

AR: In the survival results, we now specify the HRs for all studied variables, including tumor type. Thank you for the suggestion.

Line 235, Based on Figure 1, there is an opposite trend for ADC and NADC from 1996 to 2007, after that, the trends were similar even though the scale of change was not the same.

AR: Thanks for the comment; in fact, in the period 1996-2007, there was an opposite trend between ADC and NADC. However, in the 2 subsequent periods, the trend remains always opposite but no longer linear, showing a peak of NADC in 2012-2015 and a recovery of ADC in the last period. It is possible to assume that the peak of NADC is due to the implementation of cancer screening that began in those years, and that the recovery of ADC in the last period is linked to the increase in late presentation of subjects with HIV infection, which is also demonstrated by HIV/AIDS surveillance data in Italy (Notiziario Istituto Superiore Sanità; Vol 36; N° 11, Novembre 2023)

 A comment has been added to the discussion in this regard.

For Figure 4, Why were the areas of color different? What does it mean?

AR: Regarding Figure 1, please refer to the answer to the previous points. As noted, similar incidence trend in recent years may be related more to a change in the common denominator – our PLWH cohort - that would require further analysis, beyond the aim of this paper (e.g., an incident cohort study design). In Figure 4, as indicated in the legend, curve and corresponding confidence band area are colored differently for each subset of patients (grouped either by tumor type or decade of tumor diagnosis). Note that the shaded areas are semitransparent so that possible overlapping of the confidence bars could be checked.

Reviewer # 3

The result was very detailed and explicit especially the survival and disease-free survival probability.

AR: Dear reviewer, thank for appreciating our work.

Here is my question, the study highlighted that the etiology of cancer in PLWH are multifaceted, were these factors or confounding variables adjusted for in ADC and NADC among PLWH?

Line 344, conclusion needs to be moved up to line 312.

AR: Conclusion section has been fulfilled and modified.

Reviewer 3 Report

Comments and Suggestions for Authors

The result was very detailed and explicit especially the survival and disease free survival probability.

Here is my question, the study highlighted that the etiology of cancer in PLWH are multifaceted, were these factors or confounding variables adjusted for in ADC and NADC among PLWH?

Line 344, conclusion needs to be moved up to line 312.

Author Response

Reviewer # 3

The result was very detailed and explicit especially the survival and disease-free survival probability.

AR: Dear reviewer, thank for appreciating our work.

Here is my question, the study highlighted that the etiology of cancer in PLWH are multifaceted, were these factors or confounding variables adjusted for in ADC and NADC among PLWH?

Line 344, conclusion needs to be moved up to line 312.

AR: Conclusion section has been fulfilled and modified.

Round 2

Reviewer 2 Report

Comments and Suggestions for Authors

The manuscript has been enhanced. Few remaining questions:

Line 183, in the text, 289 PLWH developed cancer. There were 166 with NADC and 120 with ADC. The total was 286. How about the other 3 PLWH?

Line 181-182, Do the authors mean the number of tumors or number of PLHIV with tumors?  Were there PLWH with more than two types of cancers?  Were there PLWH with both NADC and ADC? How were these PLWH categorized?

Figure 1, What might be the reason that the incidence of NADC increased sharply from the period of 2008-2011 to 2012-2015?

Figure 3, The survival rates were better for NADC than ADC in two-year follow up but reversed 15 years follow-up. Why was that?

Also, what was the reason that the overall survival in 2 year follow up for PLWH with ADC diagnosed in 2014-2023 was lower than in 2004-2013? Similarly, for the disease-free survival in two-year follow up was also lower for PLWH with ADC in 2014-2023 than PLWH diagnosed in 2004-2013. Why?  Medicine has advanced.

Line 173, Person here years should be a typo.

Author Response

Reviewer 2:

The manuscript has been enhanced. Few remaining questions:

Line 183, in the text, 289 PLWH developed cancer. There were 166 with NADC and 120 with ADC. The total was 286. How about the other 3 PLWH?

AR: For 3 patients out of 289, it was not possible to determine the type of tumor (missing data). We now specify it in the results. Thank you for the correction.

Line 181-182, Do the authors mean the number of tumors or number of PLHIV with tumors?  Were there PLWH with more than two types of cancers?  Were there PLWH with both NADC and ADC? How were these PLWH categorized?

AR: Throughout that description, numbers refer to first tumor diagnosis. Therefore, the number of tumors analyzed coincide with the number of included patients (289). Statistics on tumor types also refer to primary tumor diagnosis. Later tumors were then included only in the disease-free survival analyses. We clarified this point adding ‘first’ or ‘primary’ when necessary. Thank you.

Figure 1, What might be the reason that the incidence of NADC increased sharply from the period of 2008-2011 to 2012-2015?

AR: The question you pose is very interesting, thank you. We have been thinking extensively about possible interpretations of Figure 1. We believe that one possible explanation for the increase in NADCs diagnoses could be a positive result and impact of the prevention campaigns and screenings implemented by our Region in those years. Another possible reason could be the aging of a large proportion of the cohort of PLWH followed up in our Centre. Unfortunately, since they are only conjectures, we decided not to explore this result much in our work.

Figure 3, The survival rates were better for NADC than ADC in two-year follow up but reversed 15 years follow-up. Why was that?

AR: To elucidate the findings illustrated in Figure 3a, please take into consideration the following factors: i) in our cohort most PLWH developed NADCs in the last decade (94 out of 289), ii) NADCs are usually diagnosed at an older age. For many cases it was therefore not possible to properly analyze long-term survival, precisely the outcome in which it is generally observed that NADCs (and subjects diagnosed at an older age) have worse prognoses. To enhance the interpretation of the data, it is advisable to employ survival curves stratified by both decade and tumor type. Notably, Figures 3c and 3d reveal that, in general, the survival rates for AIDS-Defining Cancers (ADCs) are superior to those for NADCs, aligning with previous findings in the literature. However, an intriguing exception is observed in our cohort during the last decade, where NADCs demonstrate better short-term prognoses. This anomaly could be attributed to either advancements in novel therapies or an unusually high frequency of NADC tumors, such as melanoma or cases diagnosed at an early stage, which typically have more favourable prognoses in our cohort.

Also, what was the reason that the overall survival in 2 years follow up for PLWH with ADC diagnosed in 2014-2023 was lower than in 2004-2013? Similarly, for the disease-free survival in two-year follow up was also lower for PLWH with ADC in 2014-2023 than PLWH diagnosed in 2004-2013. Why?  Medicine has advanced.

AR: What you report from the Figures is true. However, please keep in mind that both the confidence bands and p-values of the Log-rank test show that these differences are not significant but could rather be related to natural random variations in our court (randomness). In addition, it is possible to consider that during the period of 2014-2023, the diagnosis of ADC was frequently associated with very advanced HIV (CD4 < 200 cells/mmc) and often accompanied by other opportunistic infections. No further investigation was carried out into this matter. We added a comment regarding this finding in the discussion.

Line 173, Person here years should be a typo.

AR: We are sorry, but “person-years” unit is correct. It indicates the total follow-up calculated in terms of person-time. Please refer to the definition given in the methods section: “Follow-up time was calculated in terms of person-years, estimating the actual time-at-risk – in years – that all participants contributed to the study.”